# A metasomatized lithospheric mantle control on the metallogenic signature of post-subduction magmatism

David A. Holwell[1], Marco Fiorentini[2], Iain McDonald [3], Yongjun Lu[2,4], Andrea Giuliani [5], Daniel J. Smith [1], Manuel Keith [1,6] & Marek Locmelis[7]

Ore deposits are loci on Earth where energy and mass flux are greatly enhanced and focussed, acting as magnifying lenses into metal transport, fractionation and concentration mechanisms through the lithosphere. Here we show that the metallogenic architecture of the lithosphere is illuminated by the geochemical signatures of metasomatised mantle rocks and post-subduction magmatic-hydrothermal mineral systems. Our data reveal that anomalously gold and tellurium rich magmatic sulfides in mantle-derived magmas emplaced in the lower crust share a common metallogenic signature with upper crustal porphyry-epithermal ore systems. We propose that a trans-lithospheric continuum exists whereby post-subduction magmas transporting metal-rich sulfide cargoes play a fundamental role in fluxing metals into the crust from metasomatised lithospheric mantle. Therefore, ore deposits are not merely associated with isolated zones where serendipitous happenstance has produced mineralisation. Rather, they are depositional points along the mantle-to-upper crust pathway of magmas and hydrothermal fluids, synthesising the concentrated metallogenic budget available.

[1] Department of Geology, University of Leicester, University Road, Leicester LE1 7RH, UK. [2] Centre for Exploration Targeting, School of Earth Sciences, ARC Centre of Excellence for Core to Crust Fluid Systems, University of Western Australia, 35 Stirling Highway, Crawley, WA 6009, Australia. [3] School of Earth and Ocean Sciences, Cardiff University, Park Place, Cardiff CF10 3AT, UK. [4] Geological Survey of Western Australia, 100 Plain Street, East Perth, WA 6004, Australia. [5] School of Earth Sciences, University of Melbourne, Parkville, VIC 3010, Australia. [6] GeoZentrum Nordbayern, Friedrich-Alexander-Universität Erlangen-Nürnberg, 91054 Erlangen, Germany. [7] Department of Geosciences & Geological & Petroleum Engineering, Missouri University of Science and Technology, Rolla, MO 65409, USA. Correspondence and requests for materials should be addressed to D.A.H. (email: dah29@le.ac.uk)

Magmatic arcs are the factories where continental crust is built. Furthermore, they host some of the greatest metal accumulations on Earth, as giant mineral deposits of copper (Cu), gold (Au), molybdenum (Mo) and associated by-products, such as tellurium (Te) and rhenium (Re). The majority of these deposits are genetically linked to magmatic and hydrothermal processes that occur during subduction, when arc magmatism is voluminous, hydrous and typically calc-alkaline in composition[1,2]. Once active subduction ceases, the Sub Continental Lithospheric Mantle (SCLM) that was metasomatised during subduction may undergo localised partial melting[3,4]. This post-subduction process forms relatively small volume, hydrous magmas that range from high-K-calc-alkaline, through silica-saturated to silica-undersaturated alkaline compositions[1], and are henceforth referred to as alkali-enriched. These melts are genetically related and emplaced during post-collisional extension[1]. During ascent through the lithosphere, these magmas may incorporate variable amounts of crustal material[5], differentiate and locally stall at a range of depths. Rocks formed from these post-subduction magmas may host ore deposits, which generally display even more pronounced Au and Te enrichments (up to ~1 wt% Te[6]) than syn-subduction deposits[6]. Whereas the Au–Te-rich nature of these deposits is well documented[6–8], the causes of this metal enrichment, and the crustal architecture in which the deposits sit, remain poorly understood.

In this framework, a thread of mineral systems with a common alkali-enriched signature can be traced through the continental crust: from lower (>15 km) and mid-crustal hydrous alkaline intrusions, which locally contain magmatic Ni–Cu–Au–Te and platinum-group element (PGE) sulfide mineralisation[3,4,9], all the way to upper crustal (<5 km) alkali-enriched plutons, which may host magmatic–hydrothermal porphyry Cu–Au–Pd–Pt–Te and epithermal Au–Ag–Te mineralisation[10–13]. In lower crustal magmatic systems, the source of volatiles and metals is relatively well constrained as being derived largely from the underlying hydrous, metasomatised SCLM[3]. Conversely, in upper crustal porphyry–epithermal systems the scenario is more complex. Whereas the volatile content of the magmas in porphyries and, by extension, overlying epithermal systems appears to be directly related to the degree of metasomatism in the SCLM[14], the source of metals may be more variable. For example, Cu enrichment can reflect repeated cycles of fractionation, crystallisation and enrichment at the base of the crust[15], whereas Au endowment can be spatially and genetically associated with the presence of localised enriched domains in the SCLM[16] and/or the depth of emplacement[17]. Furthermore, relatively high volatile contents and oxidation states could locally suppress early sulfide segregation in post-subduction magmas[18]. This process could potentially create the conditions for a chalcophile metal-rich volatile phase to be released upon ascent, thus explaining the enrichment of Cu and Au in the shallow crustal porphyry–epithermal systems[2,13].

Questions arise as to what the ultimate source of the volatiles and metals in the SCLM is; and whether there is a common metallogenic fingerprint that links Au–Te mineral systems associated with alkali-enriched magmas emplaced at variable crustal levels to a common and traceable genetic thread. Many post-subduction geodynamic settings are characterised by significant mantle-to-crust fluxes of metals and volatiles. Here, we show that Te can act as a direct tracer of this flux, and of both the magmatic and hydrothermal processes involved in driving it. Due to its enrichment in post-subduction settings, and to its incompatible and chalcophile behaviour in magmatic and hydrothermal systems, Te can be used as a robust tracer to identify the spatial and genetic links between deep magmatic Ni–Cu–PGE–Au–Te occurrences and more evolved, shallower, alkali-enriched porphyry–epithermal Cu–Au-(PGE-Te) systems. They have never previously been considered to be linked, but we argue that they effectively form a lithospheric-scale metallogenic continuum.

## Results

**Chalcophile element signatures of post-subduction magmas and mineralisation.** During active subduction, >10% partial melting of the fertile, metasomatised asthenospheric mantle wedge leads to voluminous basaltic magmatism that produces large mafic underplating complexes[19]. These interact with the base of the crust through melting, assimilation and fractionation, and produce the largely andesitic magmas typical of arcs[1]. Conversely, post-subduction magmatism generally comprises <10% partial melting of subduction-metasomatised SCLM, and/or lower crustal melts[10]. This process produces smaller volume, hydrous, alkali-enriched melts, which may pond and crystallise at various depths in the crust[1]. The heat engine for post-subduction melting is provided either by asthenospheric upwelling due to lithospheric extension; and/or slab drop-off and delamination; and/or post-collisional thermal rebound[2]. We present here an idealised metallogenic section of the lithosphere, using a series of natural laboratories that best represent the metasomatised SCLM (mantle rocks from the Ivrea Zone, Italy, as well as xenoliths from the Kapvaal Craton, brought up in the Bultfontein kimberlite in South Africa, and from Lihir, Papua New Guinea); magmatic settings from the lower and mid crust (ultramafic pipes from the Ivrea Zone, Italy, and lamprophyric intrusions from Scotland and Australia), and shallower magmatic–hydrothermal systems, such as porphyry systems (Gangdese Belt, China; Skouries, Greece and British Columbia, Canada) and epithermal systems (Cripple Creek, Colorado, USA). Whereas these are from a range of selected global localities, they all share a common association with post-subduction magmatism.

We present new chalcophile element data together with major element compositions for magmatic and hydrothermal mineralisation within a series of trans-lithospheric Au–Te-bearing, hydrous, alkali-enriched systems, and compare them with the results from metasomatised mantle rocks and xenoliths (Fig. 1; Table 1). Chalcophile element data are generally presented on line plots normalised to chondrite or primitive mantle[20], with elements ordered left to right by decreasing compatibility (Fig. 1a). Nickel, Co, Os, Ir and Ru are variably compatible (i.e., preferentially concentrated) in the residuum of melting and in minerals that form early from a melt (i.e., olivine, Cr-spinel, Ni–Fe sulfides). Conversely, Rh, Pt, Pd, Au, Cu and Te are incompatible and enriched in the liquid phase during partial melting and melt differentiation. Variations in the slope and peaks in these profiles reflect variations in inter-metal ratios, which can be attributed to various fractionation and/or depositional processes; they are thus useful for tracking the evolution of chalcophile metal-bearing systems[20]. Increasing degrees of partial melting and crystal fractionation, both act to steepen a normalised slope by retaining compatible elements in the source or removing them from the melt during fractional crystallisation (Fig. 1a). Conversely, a negative slope indicates a source previously modified by removal of the incompatible elements through low degrees of partial melting (Fig. 1a).

Many of the rocks from which we display data contain some degree of mineralisation, as shown by enriched Ni, Cu, PGE, Au and/or Te (Table 1). Owing to the chalcophile nature of the elements shown in Fig. 1, the profiles of such samples represent the signature of the mineralisation, with the exception of the mantle rocks and some of the upper crustal porphyry samples (i.e., Gangdese belt), which are unmineralised. It is important to note the possible effects of hydrothermal alteration on these profiles as alteration and mineralisation processes may cause mobilisation and/or fractionation of the chalcophile elements. We essentially

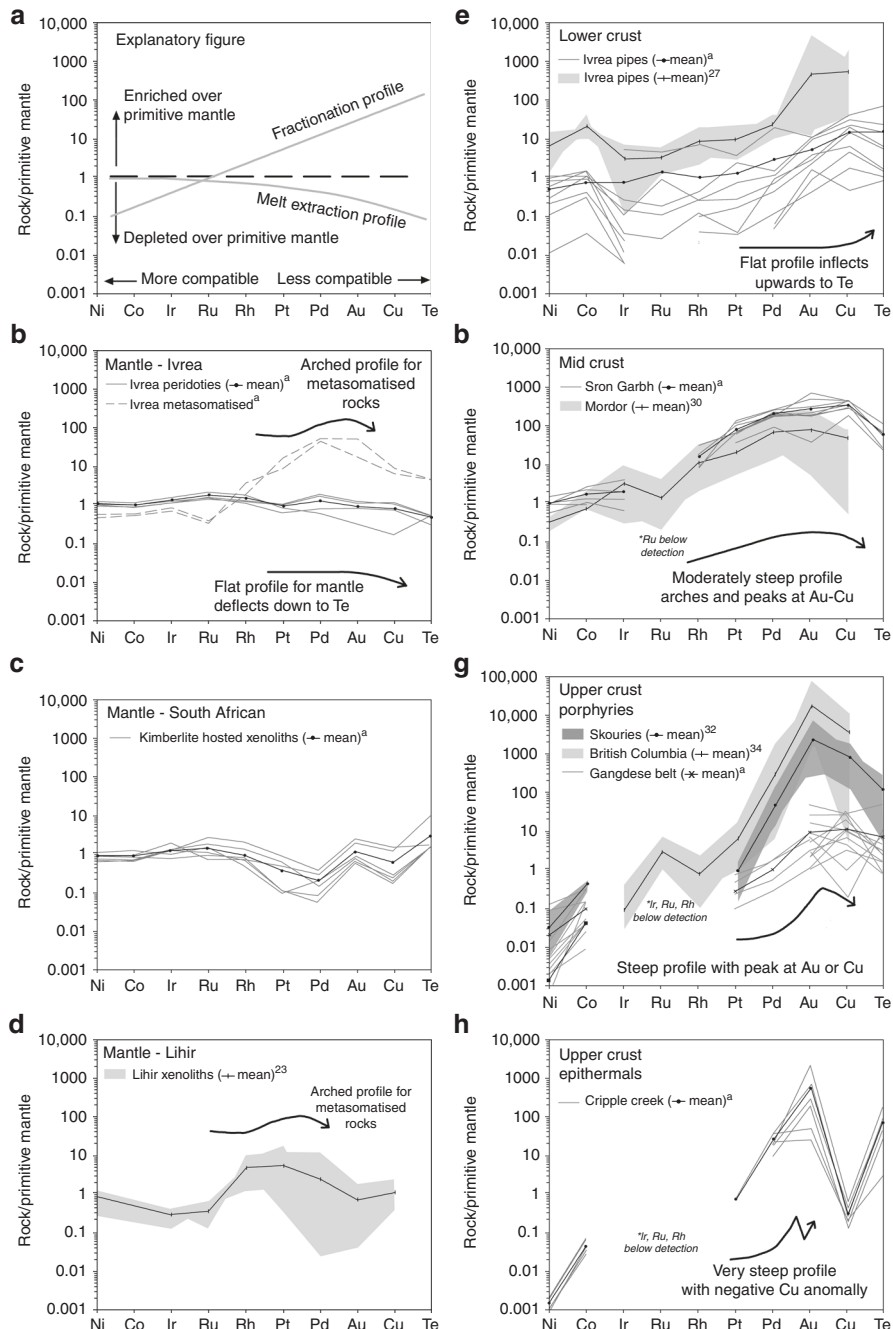

**Fig. 1** Trans-lithospheric post-subduction chalcophile element profiles. Primitive mantle-normalised chalcophile element profiles as explained in (**a**) for: **b-d** mantle rocks from the Ivrea Zone (**b**), xenoliths from the Bultfontein kimberlite (**c**) and Lihir (**d**); **e** lower crustal intrusions; **f** mid-crust intrusions; **g** upper crustal magmatic–hydrothermal porphyry systems and **h** epithermal systems. Note: "a" indicates data from this study, whereas other data are cited in reference list. Normalisation values are from Palme and O'Neill[60]

have samples that contain magmatic sulfide mineralisation, whose chalcophile element concentration profiles may be modified by hydrothermal alteration; and porphyry/epithermal mineralisation, whose chalcophile element concentration profiles are the result of hydrothermal processes. Major element compositions and loss on ignition (LOI) data are also supplied in the Supplementary Data, whereas the implications of any mineralisation/hydrothermal alteration processes are discussed below.

**Mantle**. Information about mantle melting and subsequent enrichment processes in the form of depleted and metasomatised

mantle rocks, respectively, is largely derived from petrochemical observation of rare exposed sections in orogenic massifs (e.g., Ivrea Zone, Italy[21,22]), as well as from SCLM xenoliths in volcanic rocks (e.g., Lihir[23] and kimberlite-hosted xenoliths, this study). The spinel harzburgite samples from Balmuccia in the Ivrea Zone display evidence of depletion due to melt extraction, with flat profiles showing a very gentle negative slope from Ru to Te (Fig. 1b) and very little evidence of alteration (i.e., LOI contents < 2 wt%; Supplementary Data 1). Almost undistinguishable to the depleted Balmuccia spinel peridotite is the signature of the kimberlite-hosted mantle xenoliths from South Africa, which show a profile from Ni to Pt that reflects the high degree of melt

**Table 1 New chalcophile element data generated within this study as shown in Fig. 1**

| Sample ID | Zone | Location | Rock type | Ni | Co | Ir | Ru | Rh | Pt | Pd | Au | Cu | Te |
|---|---|---|---|---|---|---|---|---|---|---|---|---|---|
| BAL1 | Mantle | Balmuccia, Italy | Spinel harzburgite | 2463.4 | 124.11 | 5.49 | 10.7 | 1.9 | 6.22 | 2.23 | 0.3 | 3.5 | 0.007 |
| BAL2 | Mantle | Balmuccia, Italy | Spinel harzburgite | 1902.6 | 99.45 | 3.94 | 7.2 | 1.3 | 6.91 | 6.2 | 1.11 | 23.7 | 0.007 |
| BAL3 | Mantle | Balmuccia, Italy | Spinel harzburgite | 2003.7 | 101.95 | 4.26 | 7.93 | 1.4 | 7.47 | 6.86 | 1.24 | 23.5 | 0.007 |
| BAL4 | Mantle | Balmuccia, Italy | Spinel harzburgite | 2165.4 | 109.75 | 4.46 | 7.49 | 1.12 | 4.55 | 2.84 | 0.78 | 16.5 | 0.004 |
| BAL6 | Mantle | Balmuccia, Italy | Lherzolite vein | 797.9 | 49.46 | 2.91 | 1.47 | 3.09 | 51.4 | 134 | 13.7 | 117.7 | 0.050 |
| BAL8 | Mantle | Balmuccia, Italy | Pyroxenite vein | 952.2 | 51.63 | 2.45 | 1.54 | 1.63 | 97.3 | 152 | 41.2 | 159.3 | 0.050 |
| XM1/142-A | Mantle | Bultfontein, RSA | Phlogopite-spinel lherzolite | 1309 | 63.0 | 4.08 | 7.48 | 1.20 | 3.87 | 0.44 | 0.56 | 3.5 | 0.017 |
| XM1/142-B | Mantle | Bultfontein, RSA | Phlogopite-spinel lherzolite | 1123 | 61.9 | 3.58 | 11.80 | 1.72 | 5.81 | 1.21 | 2.01 | 27.5 | 0.019 |
| XM1/341 | Mantle | Bultfontein, RSA | Phlogopite-spinel lherzolite | 1635 | 91.2 | 5.95 | 3.22 | 0.62 | 2.96 | 0.93 | 1.62 | 22.2 | 0.113 |
| XM1/345 | Mantle | Bultfontein, RSA | Phlogopite-spinel lherzolite | 1885 | 113.8 | 3.07 | 6.57 | 0.47 | 0.59 | 0.65 | 0.77 | 5.5 | 0.017 |
| XM1/355 | Mantle | Bultfontein, RSA | Garnet harzburgite | 1640 | 81.9 | 2.31 | 4.19 | 0.72 | 0.65 | 0.25 | 0.52 | 4.4 | 0.016 |
| XM1/422 | Mantle | Bultfontein, RSA | Spinel harzburgite | 1721 | 80.2 | 3.87 | 4.73 | 0.60 | 0.66 | 0.17 | 0.46 | 3.3 | 0.017 |
| V-LZD2 | Lower crust | Valmaggia, Italy | Amphibole-phlogopite lherzolite | 4570.3 | 255.0 | | | | 15.8 | 279.7 | 11.3 | 830 | 0.732 |
| V-LZAB | Lower crust | Valmaggia, Italy | Amphibole-phlogopite lherzolite | 2815.9 | 231.0 | | | | | 32.1 | | 1299 | 0.286 |
| V-LZD1A | Lower crust | Valmaggia, Italy | Amphibole-phlogopite lherzolite | 4080.0 | 205.1 | | | | 39.4 | 322.0 | 9.5 | 997 | 0.505 |
| V-LZD1B | Lower crust | Valmaggia, Italy | Amphibole-phlogopite lherzolite | 3109.9 | 176.3 | | | | 12.1 | 263.8 | 31.3 | 1395 | 0.297 |
| V-LZ-D1C | Lower crust | Valmaggia, Italy | Amphibole-phlogopite lherzolite | 2981.5 | 169.6 | | | | 11.3 | 230.9 | 7.5 | 718 | 0.229 |
| V-L2A | Lower crust | Valmaggia, Italy | Amphibole-phlogopite lherzolite | 2021.7 | 168.6 | | | | | 83.1 | | 544 | 0.139 |
| V-L2B | Lower crust | Valmaggia, Italy | Amphibole-phlogopite lherzolite | 881.8 | 93.8 | | | | | 94.0 | 10.5 | 740 | 0.116 |
| V-L2C | Lower crust | Valmaggia, Italy | Amphibole-phlogopite lherzolite | 727.0 | 64.8 | | | | 6.09 | 69.0 | | 400 | 0.099 |
| V-L3B | Lower crust | Valmaggia, Italy | Amphibole-phlogopite lherzolite | 516.6 | 39.4 | | | | | 63.9 | | 455 | 0.174 |
| VMG2 | Lower crust | Valmaggia, Italy | Amphibole-phlogopite lherzolite | 1183.9 | 154.3 | 0.12 | 0.13 | 0.12 | 0.27 | 3.08 | 2.55 | 415.6 | 0.070 |
| VMG5 | Lower crust | Valmaggia, Italy | Amphibole-phlogopite lherzolite | 209.70 | 34.65 | 0.02 | <0.08 | 0.02 | <0.17 | 0.17 | 0.80 | 99.40 | 0.016 |
| VMG6 | Lower crust | Valmaggia, Italy | Amphibole-phlogopite lherzolite | 421.90 | 44.93 | 0.08 | <0.08 | 0.10 | 1.18 | 1.60 | 2.28 | 321.90 | 0.060 |
| VMG7 | Lower crust | Valmaggia, Italy | Amphibole-phlogopite lherzolite | 572.80 | 116.32 | 0.04 | <0.08 | 0.04 | 0.23 | 0.38 | 0.39 | 37.20 | 0.013 |
| I7 | Lower crust | Valmaggia, Italy | Amphibole-phlogopite lherzolite | >4100 | >187 | 18.11 | 22.20 | 6.98 | 26.50 | 69.30 | 10.10 | 863.90 | 0.950 |
| I2 | Lower crust | Valmaggia, Italy | Amphibole-phlogopite lherzolite | 1975.9 | 155.26 | 0.38 | 4.12 | 0.25 | 1.82 | 2.51 | 3.64 | 129.00 | 0.018 |
| FDD1 | Lower crust | Valmaggia, Italy | Amphibole-phlogopite lherzolite | 2131.3 | 116.33 | 0.88 | 0.86 | 0.43 | 17.50 | 5.03 | 10.40 | 654.10 | 0.290 |
| FDD1A | Lower crust | Valmaggia, Italy | Amphibole-phlogopite lherzolite | 1526.0 | 98.56 | 0.47 | 0.51 | 0.28 | 5.11 | 3.94 | 9.01 | 490.40 | 0.170 |
| FDD2 | Lower crust | Valmaggia, Italy | Amphibole-phlogopite lherzolite | 21.40 | 3.80 | 0.02 | <0.08 | 0.02 | <0.17 | 0.22 | 1.42 | 9.80 | 0.011 |
| SGAQ14-13 | Mid crust | Sron Garb, Scotland | Appinite | 1602.0 | 276.1 | | | | 410.2 | 563.3 | 185.0 | 6115.1 | 0.288 |
| SGAQ15-06 | Mid crust | Sron Garb, Scotland | Appinite | 1022.8 | 100.7 | | | | 242.6 | 301.6 | 32.1 | 3777.5 | 0.268 |
| SGAQ15-07 | Mid crust | Sron Garb, Scotland | Appinite | 2715.8 | 225.7 | | | | 514.9 | 627.0 | 146.8 | 8840.9 | 0.638 |
| SGAQ16-12 | Mid crust | Sron Garb, Scotland | Appinite | 1800.0 | 132.1 | | | | 380.9 | 755.8 | 158.7 | 5783.8 | 0.768 |
| SGAQ16-13 | Mid crust | Sron Garb, Scotland | Appinite | 1720.5 | 90.9 | | | | 774.2 | 794.2 | 275.3 | 9151.8 | 1.208 |
| G318-4 | Upper crust | Gangdese, China | Amphibolite | 10 | 16 | | | | 5.4 | 6 | 6 | | 0.030 |
| BR-1 | Upper crust | Gangdese, China | Strongly altered porphyry | 8 | 5 | | | | 0.5 | | 7 | 20 | 0.110 |
| CJ-3 | Upper crust | Gangdese, China | Altered granite porphyry | 10 | 4 | | | | 0 | | 2 | 733 | 0.030 |
| BR-2 | Upper crust | Gangdese, China | Biotite monzonite porphyry | 4 | 1 | | | | 0.8 | | 5 | 461 | 0.030 |
| GJ-3 | Upper crust | Gangdese, China | Tonalite porphyry | 17 | 5 | | | | 0 | | 16 | 217 | 0.040 |
| GJ-1 | Upper crust | Gangdese, China | Granite porphyry | 4 | 3 | | | | 0 | | 2 | 263 | 0.050 |
| GJ-4 | Upper crust | Gangdese, China | Diorite | 183 | 17 | | | | 0.7 | 1 | 1 | 238 | 0.060 |
| PB-2 | Upper crust | Gangdese, China | Mafic dyke | 1 | 18 | | | | 0 | | 1 | 72 | 0.020 |
| CJ-4 | Upper crust | Gangdese, China | A-vein bearing granite porphyry | 5 | 5 | | | | 0 | | 47 | 617 | 0.670 |
| TG-3 | Upper crust | Gangdese, China | K-feldspar porphyritic tonalite | 10 | 5 | | | | 0 | | 4 | 100 | 0.060 |
| CJ-1 | Upper crust | Gangdese, China | Monzogranite porphyry | 7 | 5 | | | | 0 | | 3 | 4 | 0.170 |
| SNM-1 | Upper crust | Gangdese, China | Metamorphic gabbro with skarn veins | 27 | 36 | | | | 3.6 | 6 | 9 | 95 | 0.010 |
| TG-2 | Upper crust | Gangdese, China | K-feldspar-plagioclase porphyritic tonalite | 9 | 6 | | | | 0 | | 25 | 634 | 0.010 |
| G318-6 | Upper crust | Gangdese, China | Lamprophyre dyke | 267 | 35 | | | | 1.7 | 2 | 3 | 156 | 0.020 |
| CC-16-01.1 | Upper crust | Cripple Creek, CO | Phonolite + quartz–fluorite-telluride veins | 3.68 | 7.17 | | | | | 60.7 | 270.8 | 8.61 | 1.325 |
| CC-16-02.2 | Upper crust | Cripple Creek, CO | Phonolite + quartz–fluorite-telluride veins | 3.74 | 6.88 | | | | | 65.7 | 1933.9 | 12.90 | 3.428 |
| CC-16-03.2 | Upper crust | Cripple Creek, CO | Phonolite + quartz–fluorite-telluride veins | n.d. | 3.13 | | | | | 76.6 | 24.5 | 4.44 | 0.045 |
| CC-16-4 | Upper crust | Cripple Creek, CO | Quartz–fluorite-telluride vein | 1.88 | 3.53 | | | | 5.05 | 126.2 | 651.4 | 5.06 | 1.601 |
| VV-16-03.3 | Upper crust | Cripple Creek, CO | Tephriphonolite with telluride veins | | 2.64 | | | | | 125.6 | 45.7 | 3.10 | 0.895 |
| VV-16-07.1 | Upper crust | Cripple Creek, CO | Tephriphonolite with telluride veins | 1.51 | 4.18 | | | | | 31.3 | 166.5 | 2.65 | 0.541 |

extraction that stabilised the Kaapvaal craton in the Archean eon[24,25]. However, these xenoliths also exhibit slightly positive Au and Te anomalies (Fig. 1c), which are consistent with metasomatic addition of minor sulfides prior to kimberlite magma entrainment[26]. The source of these metasomatic agents is considered to be recycled crustal material, including sedimentary and altered oceanic crust components, which was related to ancient subduction zones at the margins of the Kaapvaal craton[26]. Other samples from the Balmuccia locality, collected from lherzolite and pyroxenite veins hosted in the spinel harzburgite,

display an enriched signature, showing a much more arched profile between Pd and Au (Fig. 1b). These compositions are comparable with those of mantle wedge xenolith samples from Lihir that are interpreted to reflect subduction metasomatism[23] (Fig. 1d).

**Lower crust**. Post-subduction alkali-enriched magmatism in the lower crust is exemplified by a suite of hydrous and carbonated alkaline ultramafic pipes that intrude the base of the exposed lower crustal section of the Ivrea Zone, which comprises a mafic underplating complex and the gneisses of the lower continental crust[3,4,27–29]. The pipes, which comprise olivine, pyroxene, amphibole, phlogopite and PGE–Au–Te-rich magmatic Fe–Ni–Cu sulfides associated with carbonate, were emplaced at ~800 °C and ~6 kbar[23]. The pipe rocks are remarkably fresh with very minor alteration. Any LOI values greater that 2 wt% (Supplementary Data 1) are related to the presence of sulfide and minor carbonate in the samples[28], which are lost on ignition as $SO_2$ and $CO_2$, respectively. As a consequence, the chalcophile profiles shown in Fig. 1 are interpreted to reflect the nature of the magmatic sulfide assemblage. Locmelis et al.[3] interpreted the pipes to have been sourced from low-degree partial melting of a metasomatised SCLM. Whole-rock chalcophile data from these pipes display relatively flat profiles from Ni to Pt, and a positive slope from Pd to Te (Fig. 1e).

**Mid crust**. Post-subduction magmatism in the mid crust is exemplified by a number of small alkali-enriched intrusions, such as Sron Garbh, Scotland[9] and the Mordor Complex, Australia[30]. Sron Garbh is a zoned intrusion with a marginal zone of hydrous lamprophyric cumulates (appinites) and a central unit of diorite. The appinites have cumulus amphibole with interstitial orthoclase, plagioclase, phlogopite, quartz, calcite and Au–Te-rich magmatic Cu–Ni–PGE sulfides[9]. Graham et al.[9] interpreted the intrusions to be low-degree partial melts of subduction-modified SCLM. The Sron Garbh intrusion is more evolved than the lower crustal Ivrea pipes, comprising amphibole-rich cumulates emplaced at >4 km[9]. The chalcophile element patterns from Sron Garbh and Mordor[30] are steeper than those from the lower crustal intrusions, with a positive slope from Ni to a peak at Au or Cu, and a negative inflection towards Te (Fig. 1f). They also display a positive Co anomaly, which has been interpreted to reflect a subduction signature inherited from their source[31].

**Upper crust**. Post-subduction alkali-enriched magmatism in the upper crust is exemplified by variably Te–Pt–Pd-enriched porphyry Cu–Au deposits from selected deposits in the Tethyan belt through Romania, Bulgaria and Greece (e.g., Skouries[32,33]), British Columbia, Canada[34], and the Gangdese belt in Tibet, China[35]. Many of the deposits in these belts are relatively Au-rich, and some may contain tellurides and display anomalously high concentrations of Pd and Pt[6,33]. We present chalcophile element profiles for the least altered porphyry igneous rocks from the Gangdese belt (i.e., 13–15 Ma-old tonalite to granite porphyries representative of fertile intrusions from the porphyry Cu deposits in the belt), along with data from ore deposits[32,34]. Samples from the Skouries and British Columbia ores have a very steep profile from Pt to Au, with a flattening or negative slope to Te (Fig. 1g). Conversely, the porphyry host rocks to the Gangdese porphyry Cu deposits have a shallower profile, but identical depletions in the more compatible elements, with a peak at Cu rather than at Au. This discrepancy is likely due to hydrothermal enrichment of the mobile Pd, Te, Cu and especially Au, from the porphyry igneous rocks into the hydrothermal deposits.

Epithermal Au–Te mineralisation associated with fluids derived from post-subduction alkali-enriched magmatism in the uppermost crust includes a number of giant Au deposits, including Cripple Creek, Colorado; Vatukoula, Fiji; and Porgera and Ladolam, Papua New Guinea[7]. Tectonically, all of these deposits formed in extensional or trans-tensional post-subduction settings. They are hosted by calc-alkaline to alkaline intrusions, with both silica-saturated and nepheline-normative compositions[36]. Whereas generally containing few base metal sulfides, they instead commonly host a telluride-rich ore assemblage with typical grades of tens to hundreds of ppm Te[7]. The chalcophile profiles of these hydrothermal systems and deposits are strongly depleted in all compatible elements (Fig. 1g, h). We present data from Cripple Creek, an Oligocene epithermal complex related to alkaline intrusions and diatremal breccias. Samples include mineralised phonolitic breccias with quartz–fluorite–telluride veins, containing disseminated pyrite and Au–Ag–Hg–tellurides alongside electrum and native silver. These samples display steep profiles similar to the porphyry deposits, with the notable difference of a strong negative Cu anomaly (Fig. 1h).

**Chalcophile element ratios**. Using the Ni/Te ratio as a proxy for the steepness of the normalised element patterns (i.e., fractionation), the new data show that there is a progressive decrease in mean bulk Ni/Te ratio of five orders of magnitude from the depleted mantle ($10^{5–6}$), through metasomatised mantle and the lower crust ($10^{4–5}$), mid crust ($10^{3–4}$), to the upper crustal porphyry ($10^{1–3}$) and uppermost crustal epithermal ($10^{0–1}$) systems (Fig. 2a). The Cu/Te ratio is relatively constant in the magmatic environments ($10^{4–5}$), but diverges significantly between the porphyry and epithermal ore systems (Fig. 2b). It is argued that this pattern reflects significant chalcopyrite precipitation at ~350 °C in the porphyry ores, as well as diminished Cu concentrations in fluids associated with the <300 °C epithermal environment[37].

## Discussion

Low volume, post-subduction magmas range from ultramafic to phonolitic in composition, with a common hydrous and variably alkaline nature[38] and a Au-(Te)-rich character[7]. In this framework, the variable geochemical signature associated with the processes of depletion due to melt extraction and subsequent refertilisation with Au, Cu and Te following subduction-related metasomatism is shown in the mantle data from the Lihir and South African xenoliths, as well as from the Balmuccia peridotite in the Ivrea Zone. Emplaced in the lower crust, the alkaline ultramafic pipes of the Ivrea Zone represent the most primitive magmas, whereas the Sron Garbh and Mordor systems represent more fractionated, lamprophyric mid-crustal intrusions. Finally, the upper-crustal porphyry–epithermal systems are associated with evolved calc-alkaline and alkaline compositions.

Although the lower-mid-crustal magmatic sulfide occurrences and the upper crustal porphyry–epithermal systems have each independently been suggested to be sourced from subduction-modified SCLM[1–4,9], they have never before been linked as parts of a continuous trans-lithospheric system. Here, we propose that these mineralised occurrences reflect a complete crustal magmatic–hydrothermal continuum (Fig. 3). The ore deposits likely represent key depositional points along the mantle to upper crust pathway taken by the magmas and hydrothermal fluids, synthesising the concentrated metallogenic budget available at that given stage. We hypothesise that the gradual variation in Ni/Te and Cu/Te with crustal depth is the result of lithosphere-scale processes, which effectively transfer and fractionate metals and

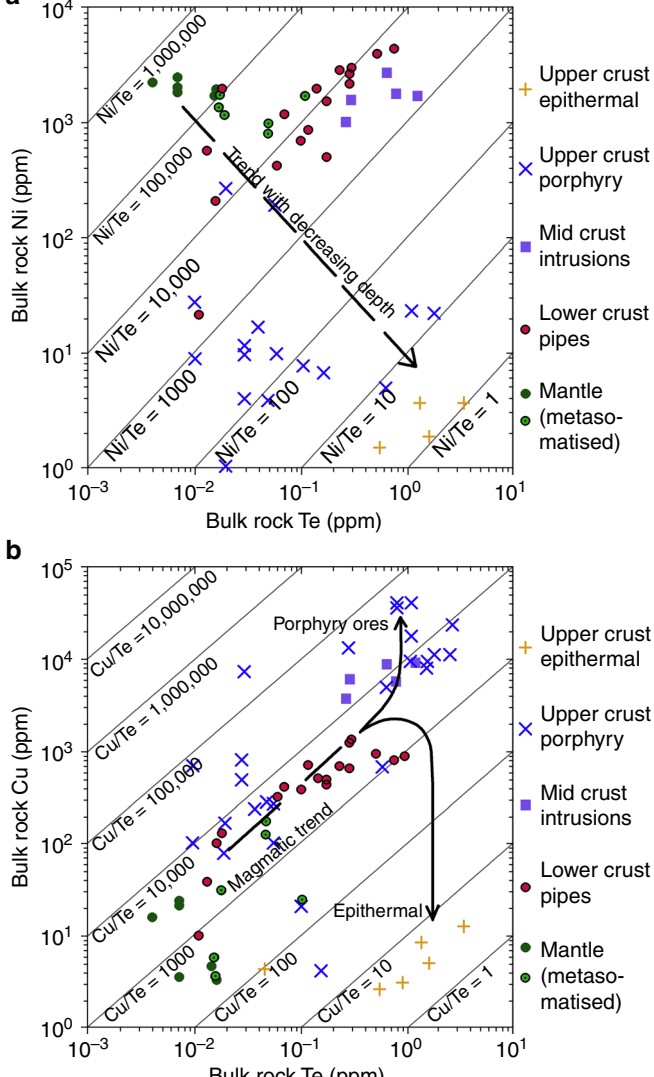

**Fig. 2** Tellurium/nickel and copper ratios through the lithosphere. **a** Plot of Te vs. Ni showing increase in Te/Ni up through the lithosphere with fractionation; and **b** plot of Te vs. Cu showing divergent behaviour in the hydrothermal environment

volatiles from the subduction-metasomatised SCLM all the way up through the crust.

In order to explain this, we use Te, and its relationship with other chalcophile elements, as a tracer of the processes of metal extraction, transport, fractionation and deposition. Tellurium has a crustal abundance of only ~5 ppb[39], but is notably enriched in volcanogenic massive sulfides and deep marine sediments, such as Fe–Mn nodules and crusts, limestone and shale (~1–200 ppm Te[7]). When subducted, these lithologies may undergo devolatilisation and/or partial melting with metasomatic mass transfer of constituent elements like Te (and Au) into the mantle wedge. Unlike other commonly used tracers of crustal cycling at subduction zones, such as radiogenic isotopes, Te is particularly useful as it will, like Au, partition into the metasomatising fluids/melts that modify the composition of the SCLM during subduction and form localised Te- (and other metal)-rich domains[40]. It is thus not surprising that magmas and mineral deposits that tap this enriched source have some of the highest Te contents of any magmatic systems. Due to its paucity in the continental crust, the Te signature of mafic magmas ascending through the crust is

only minimally affected by crustal assimilation; thus, it is inferred that the observed Te-enriched signature in post-subduction alkaline melts reflects the primary, subduction-modified mantle source.

The variable geochemical behaviour of Te at different P–T conditions provides the key constraint to support this hypothesis. At magmatic temperatures (> 1000 °C), Te behaves as a chalcophile element, like Au, Cu and the PGE. Therefore, it is readily concentrated into sulfide melts, which cool and fractionate, generally forming Pt–Pd-telluride melts at ~900 °C and then Pt–Pd-telluride minerals at <400 °C[41,42]. In hydrothermal environments, Te can be mobilised as chloride complexes at ~300 °C[40], as polytellurides in S- and $CO_2$-rich fluids[36,43], and as telluride–bismuthide melts[33]. The combination of these aspects of Te behaviour underlines its applicability to understanding mantle-to-crust fluxes of metals, as it is an ideal tracer of both magmatic and hydrothermal processes. Although Te is not generally included in routine analyses, we present here the largest and most comprehensive Te data set to date, in order to illustrate its use as a tracer alongside other chalcophile elements of magmatic–hydrothermal processes.

It is widely assumed that the vast majority of the chalcophile element budget of the mantle resides in sulfide phases[25,44]. Sulfides from anhydrous mantle domains mainly comprise Ni-rich monosulfide solid solution (mss), Cu-rich intermediate solid solution (iss), Pt–Pd-semi-metal minerals and Au phases[42]. The formation of anhydrous mafic/ultramafic magmas can involve variable degrees of partial melting, with major melting events (e.g., linked to plumes) generating >15% of mantle melting. Such large degrees of melting will consume virtually all the sulfides in the source region[45], thus effectively transferring the full incompatible chalcophile element budget to the melt. Sulfides in hydrous, metasomatised mantle domains may also be enriched in elements, such as Te and Au introduced from the subducted slab (Fig. 3a). The generation of alkali-enriched magmas from this metasomatised source in post-subduction settings involves lower degrees of partial melting (<10%) compared with the larger-scale melting that predominantly occurs during active subduction. We argue that low-degree partial melting of metasomatised SCLM during post-subduction magmatism is the trigger for transfer and concentration of metals and volatiles in magmatic arcs. A key aspect lies in the fact that relatively low-degree partial melting of previously metasomatised mantle sources preferentially concentrates S and incompatible trace metals (e.g., Cu, Au and Te) through incongruent sulfide melting[46]. This process accentuates the magnitude of the geochemical signature related to the fractionation and concentration mechanisms that are essential for ore genesis.

Incongruent melting of sulfides during low-degree partial melting of the mantle would preferentially leave some of the Ni–Os–Ir–Ru-rich mss behind, but liberate the Cu-sulfide and Au–Pt–Pd–Te phases, which have lower melting temperatures, thus producing mantle melts that are enriched in these incompatible elements[46]. Any mantle restite generated from this melting process would therefore be depleted in Cu–Au–Te relative to Ni and Ir (Fig. 1a, b), with a subsequent enrichment in the melt, as reflected in the metallogenic fingerprint of the pipes hosted in the lower continental crust (Fig. 1e). This process represents the first crucial stage in the transfer of metals and volatiles from the mantle to the crust (Fig. 3b).

Whereas the SCLM is the most likely source of metals in these systems, it can be argued that some may be sourced from the lower crust as well. For example, the generally more calc-alkaline post-subduction Gangdese porphyries have been suggested to be a product of melts from juvenile subduction-modified lower crust, based on their high Sr/Y and La/Yb ratios and unevolved isotopic

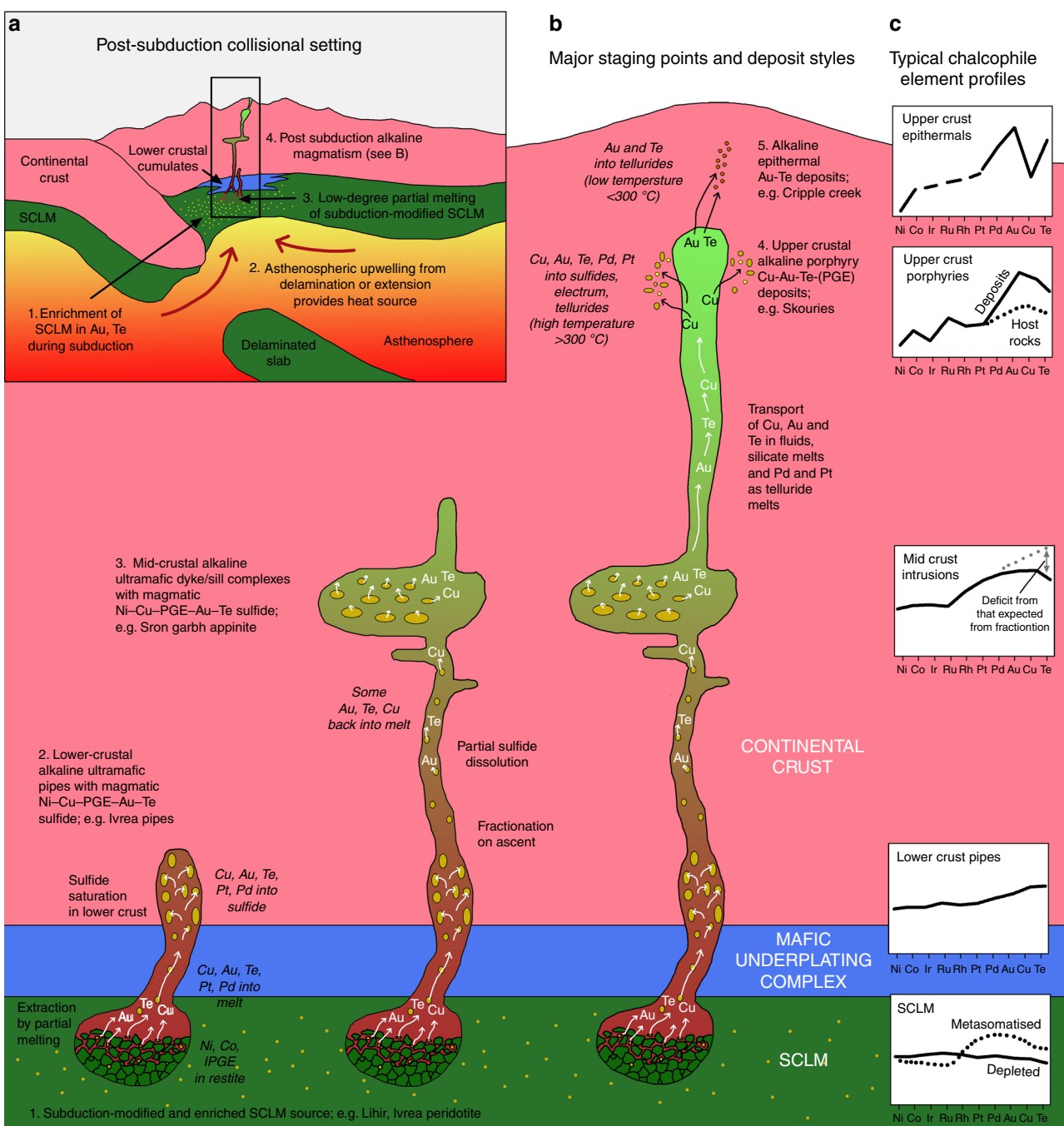

**Fig. 3** Metallogenic evolution of post-subduction magmatism. **a** Schematic representation of post-subduction magmatic setting; **b** idealised representation of hydrous alkaline magmatic systems that stage in the lower, mid or upper crust, as well as associated representative mineral deposits; **c** representative chalcophile element profile for each of the scenarios shown in **b**

compositions[47,48]. However, several studies[14,35,49] have clearly shown that lower-crust melting alone cannot provide sufficient water to these systems (>10 wt%). Post-subduction magmatism in the Gangdese belt ranges from ultrapotassic rocks[10], through shoshonites[35] and medium K-calc-alkaline[14] to high-K-calc-alkaline[14], and the ore forming high Sr/Y porphyries have been shown to be genetically related to the alkaline and shoshonitic melts from metasomatized Tibetan mantle[14,35,49]. Thus, it appears that although crustal melts may constitute the major source component in the Gangdese belt, the input of hydrous mantle melts from the metasomatised Tibetan lithospheric mantle is a small, but important requirement for the genesis of post-subduction porphyry mineralisation.

In the lower crust, most magmas will be saturated in sulfide due to the inverse relationship between S solubility and pressure[50]. Staging of a sulfide saturated system in the lower crust will produce magmatic sulfide-rich deposits, as exemplified by mineralisation hosted in the pipes of the Ivrea Zone (Fig. 3). If the magmas ascend further, the more compatible elements Ni and Co will already have been fractionated from the incompatible Cu, Au and Te by concentration into early-forming silicates like olivine, and therefore removed from the melt; this process can be traced using the varying Ni/Te ratio[51] (Fig. 2a). The resultant effect on the full chalcophile element profiles of any ore system in the mid crust will be a steepening of the normalised chalcophile element profile, as indeed observed in Fig. 1e. However, although the

profiles of the mid-crustal intrusions show a steepening, the slope flattens out and inverses at Au, Cu and Te (Fig. 1f). The observed pattern in these intrusions indicates an apparent deficit in Au, Cu and Te compared with the expected metallogenic budget predicted from incompatible element fractionation alone (Figs 1a, 3c).

Upon ascent, sulfide liquid droplets containing the bulk of the chalcophile element budget would begin to redissolve into the silicate melt due to the increase in magma S solubility[52], as the result of decreasing pressure[50]. The rate of chemical transfer is enhanced by the very hydrous nature of the system[53]. The elements with the lower partition coefficients into sulfide ($D_{sul/sil}$), such as Cu (421–1108)[55], followed by Te (1005–10,000)[54] and Au (11,200)[55], will partition into the silicate melt first, with the PGE retained in the sulfide liquid phase ($D_{sul/sil}$ of Pt ~300,000 and Pd ~500,000)[55]. As such, the silicate melt would become relatively enriched in Cu, Au and Te, whereas any remaining sulfide liquid would become relatively depleted in these elements, as seen in the sulfide-bearing mid-crust intrusions (Figs 1c, 3b). Thus, it is argued that the observed steepening of the profiles reflects the process of sulfide fractionation (also seen by Ni/Te decrease; Fig. 2a) that occurs during magma ascent in the mid crust. However, the concurrent sulfide dissolution and release of Cu, Au and Te back into the silicate melt produces a downward inflection in the profiles for these elements (Fig. 3c). The Ni/Te ratio is therefore higher than expected due to this effect, which may be the reason why samples from the lower- and mid-crust plot much closer to each other in Fig. 2a. Sulfide dissolution represents a key process in the evolution of such magmas and leads to the priming of the melts for subsequent Cu–Au–Te ore genesis at upper crustal levels.

However, the characteristic enrichment of Pd, and to a lesser extent, Pt in some post-subduction porphyry systems cannot be explained by this mechanism alone as it would imply that these elements, which have the highest $D_{sul/sil}$ values, ought to remain in the sulfide melt. Therefore, an additional factor to account for the observed enrichment is likely to be the role of telluride melts at temperatures below 900 °C[33,42,56]. It is likely that some Te in the sulfide liquid at this temperature will form a telluride melt that is incompatible in crystalline sulfide, but into which Pd and Pt are highly compatible. Therefore, while sulfide dissolution may be occurring at this stage, there may also be a co-existing Pd–Pt–telluride melt, which will remain liquid to low temperatures (~400 °C)[41]. Thus, while in mid- to upper crustal systems Pt and Pd are likely to be transported by telluride melts, Cu and Au may be redissolved into the silicate melt. Telluride melts have been suggested to be the reason for the Pd–Pt–Te enrichment in the Skouries deposit[33]; and we propose this to be a viable mechanism for enriching some upper crustal deposits in Pd and Pt, but not in the other PGE.

At mid- to upper crustal levels, the evolved, likely sulfide-undersaturated magmas[50] have the potential to transport a heavily fractionated metal budget, depleted in Ni, Co, Os, Ir, Ru, Rh, but enriched in Cu, Te and Au (and possibly also Pt and Pd in the presence of telluride melts; Fig. 1f, g). Our data from the non-mineralised, least altered Gangdese host rocks support this hypothesis, with a fractionated profile shown in Fig. 1g that is comparable with the mid-crust profiles. The metals may be enriched in the silicate melt following the priming process described above, be transported in a volatile component as, for example, chloride complexes, or potentially as telluride melts[33,41]. Subsequent ore formation from these melts, such as magmatic–hydrothermal Cu–Au–(Te–Pd) deposits in the upper crust (e.g., Skouries), displays heavily fractionated, steep profiles (Fig. 1g). The key concept here is that the deep magmatic processes fractionate metals to enrich the porphyry magmas with a metal cargo that is preferentially enriched in Cu–Au–Te(–Pd–Pt). Most importantly, these processes are limited in space and time.

The contrasting metal ratios between porphyry and epithermal deposits (Fig. 1d) likely reflect hydrothermal processes that fractionated Cu from Au and Te (Fig. 2b) between the higher temperature (>300 °C) porphyry and lower temperature (<300 °C) epithermal environments[40] (Fig. 3b). Knowledge about the behaviour of Te at the porphyry–epithermal transition is limited, but fluid boiling is known to be one of the prevalent processes of ore formation in these environments. Fluid boiling could lead to depositon of Cu in porphyry environments, whereas Te partitions into vapour and precipitates at shallow crustal levels in Au-rich zones under epithermal conditions[13,57].

Alkali-enriched magmas emplaced at the base of the continental crust have the potential to ascend further, transporting mantle-derived volatiles and metals through the continental lithosphere all the way to uppermost crustal settings. Deep magmatic processes have a profound effect on the metal budgets of melts that reach the upper crust, and thus play a key role in controlling the abundance of the characteristic metals that are enriched in post-subduction settings. The proposed metallogenic continuum most likely operates partially or completely in different localities, depending on the pre-existing tectonic architecture. The ore deposits represent key staging points for the system as a whole: where clusters of epithermal Au–Te deposits occur, the system has progressed to very shallow crustal depths; in other cases, aborted systems may stall at any point in the crust. Post-subduction settings contain magmatic systems that, due to their small volume and enriched metal source, magnify lithospheric metal and volatile concentration and transfer through a continuous (or staged) ascent from the mantle to the upper crust. However, the widely documented association of Te–Au deposits with alkali-enriched magmatism is not mutually causative. Rather, the alkali-enriched nature of the igneous rocks and the Te–Au-rich signature of the metal budget are separate functions of the same broader process: low-degree partial melting of a hydrous, subduction-metasomatised mantle source.

## Methods

**Samples**. Chalcophile element abundances were determined in this study for a total of 54 samples from selected localities: the Balmuccia mantle peridotite and one of the lower crustal alkaline ultramafic pipes (the Valmaggia pipe) in the Ivrea Zone, Italy; peridotite xenoliths from the Bultfontein kimberlite (Kimberley, South Africa); the mid-crust Sron Garbh appinite intrusion, Scotland; Gangdese porphyry rocks in southern Tibet, China and the upper crustal Cripple Creek deposit, Colorado, USA. Major and chalcophile trace element data are shown in the Supplementary Data. In addition, to complement our new data, we utilise published data on metasomatised mantle xenoliths from Lihir[23], Papua New Guinea; other alkaline lower crustal pipes from the Ivrea Zone[27]; the mid-crust alkaline Mordor Intrusion, Australia[30] and the upper crustal Skouries porphyry deposit, Greece[32], and deposits in British Columbia[34].

**Bulk rock geochemistry**. A suite of samples from the Ivrea Zone and peridotite xenoliths from the Bultfontein kimberlite were analysed by ICP-MS analysis at Geoscience Laboratories (GeoLabs, Ontario Geological Survey, Sudbury, Canada), and a separate suite from Valmaggia was analysed by XRF at the University of Leicester, UK. Platinum-group element data and gold concentrations were obtained using the conventional nickel-sulfide fire assay pre-concentration technique. An aqua regia extraction step was necessary for the accurate determination of tellurium values. Analytical details for all the techniques utilised to generate the PGE data in this study, including sample preparation, accuracy and precision information for a diverse range of standards and internal reference samples, have been described by Barnes and Fiorentini[58] and Fiorentini et al.[59], who also provide information on inter-laboratory reproducibility, sample homogeneity and reproducibility of determinations on standard materials.

Bulk rock major element compositions of samples from Cripple Creek samples were determined using a PANalytical Axios-Advanced XRF spectrometre, operating with PANalytical SuperQ software at the University of Leicester on pressed powder pellets. Full PGE (Pt, Pd, Rh, Ru, Ir, Os) and Au analyses of Sron Garbh samples were undertaken using 30 g of samples by fire assay with nickel-sulfide collection and neutron activation analysis at ALS Geochemistry

(PGM-NAA26) as reported in Graham et al.[9]. No major element analysis was performed on these rocks. Tellurium contents of samples from Sron Garbh and Te, Au, Pt and Pd contents of samples from Cripple Creek were analysed by aqua regia digest followed by ICP-MS finish at Cardiff University, UK. Masses close to 0.5 g of milled and dried powder were digested in 5 ml of concentrated aqua regia (three parts HCl:one part $HNO_3$) inside a sealed 15-ml capacity screw-top Teflon vial on a hotplate at ~150 °C for 16–18 h. Samples were allowed to cool and settle before 0.5 ml of digest solution was extracted using a pipette and diluted to 5 ml with 18.2 molar deionised water. Diluted samples were analysed on a Thermo iCAP RQ ICP-MS. Highly chalcophile elements such as those that are associated with sulfide minerals are assumed to be close to 100% extracted by aqua regia.

Bulk rock compositions of Gangdese samples were determined at ALS Minerals in Perth, Western Australia. Major elements (Si, Al, Fe, Ca, Mg, Na, K, Ti, Mn, P, LOI) were determined by lithium metaborate fusion with ICP-AES (ALS code ME-ICP06). Trace- element compositions were measured by lithium borate fusion with ICP-MS with base metals (Ag, Cd, Co, Cu, Li, Mo, Ni, Pb, Zn) and Sc determined by four-acid digestion with ICP-AES (ALS code ME-4ACD81). Volatile trace elements (As, Bi, Hg, In, Re, Sb, Se, Te) were measured by aqua regia with ICP-MS (ALS code ME-MS42). Precious metals (Au, Pt, Pd) were analysed using 30 g of pulp by fire assay with Pb collection and ICP-MS finish (ALS code PGM-MS23).

## Data availability

All correspondence and material requests should be addressed to D.A.H. dah29@le.ac.uk.

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

## Acknowledgements

This work is funded by NERC SoS Consortium grant NE/M010848/1 and NE/M011615/1 "TeaSe: tellurium and selenium cycling and supply", and NERC grant NE/P017053/1 and NE/P017312/1 "FAMOS: from arc magmas to ores"; both awarded to the University of Leicester and Cardiff University, respectively. The study was also funded by the Australian Research Council Centre of Excellence for Core to Crust Fluid Systems (CE11E0070). Y.L. acknowledges a Tibet pilot project from CCFS and publishes with permission of the Executive Director of GSWA. This is contribution 1356 from the ARC Centre of Excellence for Core to Crust Fluid Systems (www.CCFS.mq.edu.au).

## Author contributions

D.A.H. contributed the bulk of the writing and generated data from Sron Garbh and the Ivrea Zone. M.F. contributed to the writing of the paper and provided data from the Ivrea Zone and South Africa. I.M. generated data for the Cripple Creek samples and contributed to writing of the paper. Y.L. generated data for the Gangdese samples and contributed to editing the paper. D.J.S. and M.K. contributed to editing the paper and provided samples from Cripple Creek. A.G. contributed to editing the paper and provided data from South Africa. M.L. contributed to editing the paper and provided data from the Ivrea Zone.

## Additional information

**Competing interests:** The authors declare no competing interests.

