## [Peer Review File · Nature Communications]

Editorial Note: This manuscript has been previously reviewed at another journal that is not operating a transparent peer review scheme. This document only contains reviewer comments and rebuttal letters for versions considered at Nature Communications .

Reviewers' comments:

Reviewer #1 (Remarks to the Author):

The metallogenic DNA of post-subduction magmatism

The paper discusses an interesting concept for the use of Te and other elements as geochemical tracers for processes in post-subduction magmatic and magmatic hydrothermal systems. The conceptual approach and the resulting suggestions for connections between deep magmatic processes and shallow magmatic-hydrothermal systems are worthy of publication.

Unfortunately, the new data are relatively sparse and are derived from disparate environments where the connection to post-subduction processes is at least uncertain. In addition, the processes that generate metal concentrations at the site of deposition, the ore deposits, receive limited attention and it is not clear why metal ratios at shallow levels should reflect deep magmatic processes. The paper acknowledges these deficiencies to some extent but contains some sweeping statements that are not clearly justified. If the conceptual arguments can be identified more clearly, and supported by the data where possible, the paper should be published. The paper contains a number of errors and typos listed below.

Title: I understand the colloquial use of "DNA", but am not comfortable with it figuratively, or obviously, literally. Given the breadth of Nature readers, it could be confusing, but I leave that call for the Editors.

Line 53: Discussion of Au endowment: Many factors may influence endowment at the deposit scale, possibly the most important being the depth of emplacement; e.g., see papers on the Kislidag porphyry Au deposits by Baker et al., and similar suggestions for the Maricunga District (the latter are subduction-related calc-alkaline systems, but have similarities to the post-subduction alkaline Kislidag deposits). Key point - the depositional setting is very important for metal budgets.

Line 86: Te may be an important pathfinder element, and the focus of this discussion, but it is hardly the most important element in this list of elements – should be Cu-Au (Te-Ni-PGS).

Line 93: Replace "form" with crystallize

Line 107, 108-109, 111: It's not just alteration - it is the hydrothermal processes at the site of deposition; alteration is indicative of these processes.

Line 110: Delete "and"

Line 128: What does "low" mean? Replace by limited?

Line 130-132: If the magmatic sulfides contain all of the chalcophile elements, element ratios and profiles will not be influenced by LOI. If hydrothermal processes indicated by alteration introduce secondary sulfides such as pyrite with contained trace elements, then primary trace element ratios might be disturbed.

Line 148-9: Misleading element list again. These are Cu-Au deposits from an economic perspective - i.e., what defines them as deposits. They show geochemical enrichment in Te, Pd and Pt, and in some cases, these elements are recovered by smelters but they do not define the "deposit". Refer to them as "Te-Pt-Pd enriched porphyry Cu-Au deposits".

Line 150: Not aware of any tellurides reported from alkaline porphyry deposits in British Columbia.

Line 158-161: Awkward sentences – rewrite as on simple sentence.

Line 176: A stretch to suggest that these all have Te-Au-rich character based on the limited data.

Line 191: Cu/Au can also reflect low temperature (<300C) processes – far more understanding and data.

Line 245: Not all lower crustal magmas will be saturated with respect to S – and certainly not "supersaturated". Depends on composition and S content as well as pressure.

Line 254: Not sure that we have enough data or understanding to say that this paradoxical.

Line 269: Missing "is" - ... this is the reason...

Line 270-272: Unsupported over-statement. This may be an important process but not proven as a

prerequisite for ore genesis. Note again these are not Cu-Au-Te ores – they may be Te-enriched Cu-Au ores.

Line 285: Again – no reported tellurides in deposits in British Columbia – even the Pd-rich deposits.

Line 296: In porphyry Cu deposits, Au usually occurs in the Cu sulfides so why would sulfides fractionate Cu from Au?

Line 310: “full maturity” – meaning? Replace with highest/shallowest level of formation?

Line 313-314: Evidence for this being the largest flux of Te? Is this in terms of total Te – in which case what data do we have to support this statement, of rate of movement?

Line 341: Samples were analyzed – not “determined”

Line 487 – Fig 2B: More data than in A; Geochemical or bulk (ore grade) Cu values? Bulk rock presumably reflects sulfide content? The lack of Cu in epithermal is usually explained by a switch from Cl to HS- complexing of gold below 300C.

Reviewer #2 (Remarks to the Author):

Summary

The manuscript presents a variation of Ni/Te and Cu/Te ratios for tracking the processes through the lithosphere in post-subduction magmatism that may result in the concentration metals and volatiles in ore systems. This paper utilizes an accepted approach to evaluate these elements suite and suggests processes that operate at different levels from the mantle to crust, trying to explain changing patterns. The result is an original manuscript which will be of general interest to those working on the metallogeny of post-subduction environment, and related research in petrology and geodynamics, however, it needs more work to demonstrate it.

The manuscript has several flaws:

- i) However, as for the samples and data with very limited from post-subduction metallogenesis - only present 4 different localities with 4 mantle, middle crust rocks to build a globe model.
- ii) These samples do not describe petrology observation and geochemical data present. It surprised that definition of the rocks derived from mantle or lower crust. Authors emphasis that metasomatized mantle contribute volatiles to mineralisation, it need present geochemistry to define metasomatized peridotites, at least, it should be given number in the main text.
- iii) Authors focus on post-subduction samples, but do not collected these from one ore systems or same metallogenic belt, or two postsubductional Cu-Au deposit.
- iv) The question from previous referee's still present. The post-subduction ore-forming magmas do not only come from SCLM. The lower crust is a common magmatic source. Take the Gangdese belt in southern Tibet (the most-typical post-subduction setting) as the example, the ore causative intrusions are calc-alkaline to high K calc-alkaline, not alkaline. These intrusions have high Sr/Y and La/Y ratios, and are derived from partial melting of lower crust.
- v) The data of Table 1 should give reader more detail geochemical information, at least the MgO and SiO₂ should be given.

Lines 43-45: As previous referee's mentioned, for example, the post-subduction Tibetan crust is much thick than 15-5Km.

Lines 47-50: Gangdese post-collisional magmatism which host porphyry-epithermal systems is derived from lower crust. How to distinguish or define lower crustal magmatic systems and upper crust? Should focus on the its source or shallow magma chamber?

Lines 64-65: not only post-subduction geodynamic settings are characterised by an enormous flux of metals and volatiles, it also should be enriched in metals and volatiles in the ongoing-subduction

arc.

Lines 73-78: This is not true, post-subduction magmatism is not generally comprised of <10% partial melting of subduction-metasomatised SCLM, on the contrast, widespread high Sr/Y and La/Yb ore-forming magmatism are derived from partial melting of lower crust.

Line 114-118: should define the information about metasomatised mantle and some geochemical evidence of subduction-related metasomatism.

Line 228-233: "We argue that low-degree partial melting of metasomatised SCLM during post-subduction magmatism is the trigger for transfer and concentration of metals and volatiles in magmatic arcs..." this needs more detailed evidence, Conversely, there is more work showing that metals came from lower crust.

Response to review:

NCOMMS-18-4584440A R1

The metallogenic DNA of post-subduction magmatism

(changed to “The metallogenic fingerprint of post-subduction magmatism” for R2)

By Holwell et al.

We appreciate the opportunity to revise and resubmit our manuscript to Nature Communications. We were pleased with the reviews which were constructive, and very largely positive, with a message that our paper is of interest, should be published, but with some concerns to be addressed before it is. We have taken on board these comments, and have addressed them all; our responses are set out in bold to each of the editor’s and reviewers’ points. We feel that the manuscript is much stronger now because of the peer review process.

Reviewer #1 (Remarks to the Author):

The metallogenic DNA of post-subduction magmatism

The paper discusses an interesting concept for the use of Te and other elements as geochemical tracers for processes in post-subduction magmatic and magmatic hydrothermal systems. The conceptual approach and the resulting suggestions for connections between deep magmatic processes and shallow magmatic-hydrothermal systems are worthy of publication.

We are pleased to see this recommendation.

Unfortunately, the new data are relatively sparse and are derived from disparate environments where the connection to post-subduction processes is at least uncertain.

We appreciate the reviewer’s comment. We have added a significant amount of new data (i.e. 54 new samples original to this study instead of 31). Together with selected literature data, we present here the biggest and most comprehensive Te dataset to date (data from 105 samples), in order to illustrate its use as a tracer alongside other chalcophile elements of magmatic-hydrothermal processes. The dataset includes new data from mantle rocks (both depleted mantle and mantle affected by subduction-related metasomatism) as well as porphyries from the Gangdese belt in Tibet, which represent the signature associated with the least altered fertile rocks as a useful comparison to the ore deposit data shown from the literature. The expanded Te

dataset strengthens the proposed model by now including two additional settings in our mantle-to-upper-crust continuum that were previously covered only using literature data. The new data and associated interpretations have been provided by Andrea Giuliani and Yongjun Lu who have joined the author list.

However, we respectfully disagree that the connection of some of the natural laboratories selected for our study to post-subduction processes is uncertain. In fact, it is argued that all the crustal examples have very strong links to post-subduction settings. This point is now emphasised by references in the individual sections that describe each of the sample suites.

In addition, the processes that generate metal concentrations at the site of deposition, the ore deposits, receive limited attention and it is not clear why metal ratios at shallow levels should reflect deep magmatic processes. The paper acknowledges these deficiencies to some extent but contains some sweeping statements that are not clearly justified. If the conceptual arguments can be identified more clearly, and supported by the data where possible, the paper should be published.

This is an excellent comment! With regards to linking with deep magmatic processes, the key concept is the idea that deep magmatic processes occurring in the lower to mid crust ‘prime’ the magmas that reach upper crustal levels with a specific metal signature. This would be a likely mechanism for generating relatively Cu-Au-Te melts, which are spatially and temporally constrained to post-subduction settings. We now include the following paragraph (lines 348-359 of revised ms):

*“At mid to upper crustal levels, the evolved, likely sulfide-undersaturated magmas⁵¹ have the potential to transport a heavily fractionated metal budget, depleted in Ni, Co, Os, Ir, Ru, Rh, but fertile in Cu, Te and Au (and possibly also Pt and Pd in the presence of telluride melts; Fig. 1C, D). Our data from the non-mineralised, least altered Gangdese host rocks support this, with a fractionated profile shown in Figure 1G that is comparable to the mid crust profiles. The metals may be enriched in the silicate melt following the priming process described above, be transported in a volatile component as, for example, chloride complexes, or potentially as telluride melts^{33,41}. Subsequent ore formation from these melts, such as magmatic-hydrothermal Cu-Au-(Te-Pd) deposits in the upper crust (e.g., Skouries), display heavily fractionated, steep profiles (Fig. 1D). **The key concept here is that the deep magmatic processes fractionate metals to enrich the porphyry magmas with a metal cargo that is preferentially enriched in Cu-Au-Te(-Pd-Pt).** Most importantly, these processes are limited in space and time.”*

As this message may have not come through in the former draft as clearly as we envisaged, we have emphasised it in the paragraph quoted above in the response to the editor's comments. The main point is in line 356-358 (bold above). In addition, we are now able to add some additional supporting evidence to this hypothesis with the addition of our samples from the Gangdese belt, which show the composition of the ‘parental’ magmas from which upper crustal porphyries are derived, and allow comparison with the mineralised rocks (Line 190-194 of revised ms).

The paper contains a number of errors and typos listed below.

Title: I understand the colloquial use of “DNA”, but am not comfortable with it figuratively, or

obviously, literally. Given the breadth of Nature readers, it could be confusing, but I leave that call for the Editors.

We have removed this from the title of the paper, and have changed it throughout the manuscript to 'fingerprint'.

Line 53: Discussion of Au endowment: Many factors may influence endowment at the deposit scale, possibly the most important being the depth of emplacement; e.g., see papers on the Kislidag porphyry Au deposits by Baker et al., and similar suggestions for the Maricunga District (the latter are subduction-related calc-alkaline systems, but have similarities to the post-subduction alkaline Kislidag deposits). Key point - the depositional setting is very important for metal budgets.

Point well taken, and this is now incorporated into this section with a reference to Baker et al. 2016.

Line 86: Te may be an important pathfinder element, and the focus of this discussion, but it is hardly the most important element in this list of elements – should be Cu-Au (Te-Ni-PGS).

Point well taken. We have rearranged the element lists to reflect the relative abundance and, therefore, importance. Nickel is however not present in detectable concentrations in the porphyries so we have left it out of this particular reference.

Line 93: Replace “form” with crystallize

Done.

Line 107, 108-109, 111: It's not just alteration - it is the hydrothermal processes at the site of deposition; alteration is indicative of these processes.

Agreed. The focus on alteration here was in response to one of the earlier reviews who suggested that we needed to show how fresh these rocks are. We have reworded this now to reflect the hydrothermal processes of mineralisation as well as alteration (line 129-136).

Line 110: Delete “and”

Done.

Line 128: What does “low” mean? Replace by limited?

Done.

Line 130-132: If the magmatic sulfides contain all of the chalcophile elements, element ratios and profiles will not be influenced by LOI. If hydrothermal processes indicated by alteration introduce secondary sulfides such as pyrite with contained trace elements, then primary trace element ratios might be disturbed.

True, but there are no significant secondary processes here and we cite the references that say so (e.g. line 126). Therefore, our comment about the ratios being representative of the magmatic sulfides still stands. As a result, this part has not been changed.

Line 148-9: Misleading element list again. These are Cu-Au deposits from an economic perspective - i.e., what defines them as deposits. They show geochemical enrichment in Te, Pd and Pt, and in some cases, these elements are recovered by smelters but they do not define the "deposit". Refer to them as "Te-Pt-Pd enriched porphyry Cu-Au deposits".

We have modified the text to accommodate the reviewer's suggestion.

Line 150: Not aware of any tellurides reported from alkaline porphyry deposits in British Columbia.

Point well taken. We have made the phrasing "variably" Te-Pt-Pd enriched Cu-Au porphyries. It may be that tellurides have not been identified, but this does not preclude their presence. They are rare, even where they are 'enriched'. Line 182 and 185 make this clearer.

Line 158-161: Awkward sentences – rewrite as on simple sentence.

Done.

Line 176: A stretch to suggest that these all have Te-Au-rich character based on the limited data.

We have accepted the reviewer's suggestion and toned down the sentence to read "Au(-Te)-rich character". The Au-Te association really does seem to be a consistent relationship in the data we have and also certainly in the epithermal deposits.

Line 191: Cu/Au can also reflect low temperature (<300C) processes – far more understanding and data.

This appears to be a misunderstanding because we agree with the reviewer's view and the manuscript indeed includes in lines 215-217:

It is argued that this pattern reflects significant chalcopyrite precipitation at ~350°C in the porphyry ores, as well as diminished Cu concentrations in fluids associated with the <300°C epithermal environment³⁷

Line 245: Not all lower crustal magmas will be saturated with respect to S – and certainly not "supersaturated". Depends on composition and S content as well as pressure.

We agree with the reviewer and have rephrased the text accordingly: This has been rephrased in lines 305-308 and 3118-320.

Line 254: Not sure that we have enough data or understanding to say that this is paradoxical.

Deleted.

Line 269: Missing "is" - ... this is the reason...

Done.

Line 270-272: Unsupported over-statement. This may be an important process but not proven as a prerequisite for ore genesis. Note again these are not Cu-Au-Te ores – they may be Te-enriched Cu-Au ores.

Point well taken. This sentence has been rephrased in line 332.

Line 285: Again – no reported tellurides in deposits in British Columbia – even the Pd-rich deposits.

Ok, point taken. This statement is a general in its nature, and references Skouries specifically. Line 345-6.

Line 296: In porphyry Cu deposits, Au usually occurs in the Cu sulfides so why would sulfides fractionate Cu from Au?

We appreciate the reviewer's concern; however, this hypothesis is taken from the study of (Grundler et al reference cited). We agree that some Au will be probably partitioned into the sulfides. On the other hand, it is also well established that Au can be transported to lower temperatures. We have rephrased this statement to avoid the incorrect assumption no Au goes into sulfide (lines 362).

Line 310: "full maturity" – meaning? Replace with highest/shallowest level of formation?

Done.

Line 313-314: Evidence for this being the largest flux of Te? Is this in terms of total Te – in which case what data do we have to support this statement, of rate of movement?

We have removed this statement.

Line 341: Samples were analyzed – not "determined"

Rephrased.

Line 487 – Fig 2B: More data than in A; Geochemical or bulk (ore grade) Cu values? Bulk rock presumably reflects sulfide content? The lack of Cu in epithermal is usually explained by a switch from Cl to HS- complexing of gold below 300C.

The dataset reported in figures 1 and 2 is not the same as some results do not have both Cu and Ni. This is because Ni is below detection in some of the porphyry/epithermal deposits.

Reviewer #2 (Remarks to the Author):

Summary

The manuscript presents a variation of Ni/Te and Cu/Te ratios for tracking the processes through the lithosphere in post-subduction magmatism that may result in the concentration metals and volatiles in ore systems. This paper utilizes an accepted approach to evaluate these elements suite and suggests processes that operate at different levels from the mantle to crust, trying to explain changing patterns. The result is an original manuscript which will be of general interest to those

working on the metallogeny of post-subduction environment, and related research in petrology and geodynamics, however, it needs more work to demonstrate it.

We are pleased with this encouraging overview, and take note of the suggestions which we feel have strengthened the paper significantly.

The manuscript has several flaws:

i) However, as for the samples and data with very limited from post-subduction metallogenesis - only present 4 different localities with 4 mantle, middle crust rocks to build a globe model.

Point well taken. We have added a significant amount of new data (now 54 samples instead of 31) as described above which, together with the literature data we use, now forms what we feel is a much more robust dataset to interrogate. This is, however, the biggest dataset of its kind that contains Te, which is generally not analysed for in routine analyses. As a result, we feel that we have a very interesting story with these data (bolstered by the extra samples) that is appropriate for such a paper.

ii) These samples do not describe petrology observation and geochemical data present. It surprised that definition of the rocks derived from mantle or lower crust. Authors emphasis that metasomatized mantle contribute volatiles to mineralisation, it need present geochemistry to define metasomatized peridotites, at least, it should be given number in the main text.

Point well taken. We have added metasomatised mantle rocks to our dataset, which include sulfide-bearing peridotite xenoliths from the Kaapvaal craton, as well as lherzolite and pyroxenite rocks from the Balmuccia peridotite in the Ivrea Zone (Italy). In each section we now describe the petrology, with references to more detailed descriptions.

iii) Authors focus on post-subduction samples, but do not collected these from one ore systems or same metallogenical belt, or two postsubductional Cu-Au deposit.

Excellent comment! As we have mentioned above, this revision now includes almost twice as many new data in relation to the previous version. To address the important concern of this reviewer, we have included results from the Gangdese porphyry system, which represent host rocks rather than mineralised rocks. These data now provide a useful comparison that still shows the diagnostic fractionation with decreasing lithospheric depth that is at the core of the proposed model. This varying signature is still visible without the magnification that is related to ore forming mechanisms (as we say in line 379 at the end of the manuscript).

iv) The question from previous referee's still present. The post-subduction ore-forming magmas do not only come from SCLM. The lower crust is a common magmatic source. Take the Gangdese belt in southern Tibet (the most-typical post-subduction setting) as the example, the ore causative intrusions are calc-alkaline to high K calc-alkaline, not alkaline. These intrusions have high Sr/Y and La/Yb ratios, and are derived from partial melting of lower crust.

Point well taken. We not only added new data from the Gangdese belt, but also a paragraph under 'Magmas and metal sources' that addresses the origin of high Sr/Y rocks in the Gangdese belt (Line 274-282). We point to a number of recent studies that show that although crustal melts are

important in this region, a mantle input is also required. Thus, we argue that our assertion that the SCLM plays a key role is still valid.

v) The data of Table 1 should give reader more detail geochemical information, at least the MgO and SiO₂ should be given.

Thanks for the comment. We don't feel this is necessary as all of the bulk rock data are in the Supplementary Table. We have added a column in Table 1 saying what the rock type is. This effectively addresses the issue by showing the composition of the rock, but also adds further information, without detracting from the main data that the table shows (i.e. the chalcophile element data that the paper focusses on).

Lines 43-45: As previous referee's mentioned, for example, the post-subduction Tibetan crust is much thicker than 15-5Km.

Point well taken. Edited to reflect this (line 61).

Lines 47-50: Gangdese post-collisional magmatism which hosts porphyry-epithermal systems is derived from lower crust. How to distinguish or define lower crustal magmatic systems and upper crust? Should focus on its source or shallow magma chamber?

Thanks for the comment. The origin of post-collisional magmatism in the Gangdese belt is debated. Whilst there is undoubtedly some crustal component, there is also a body of work that shows that some mantle input is required (Lu et al., 2015; Yang et al., 2015; Sun et al., 2018). We have added a paragraph in the discussion under Magmas and Metal Sources to reflect this (Line 296-302 of revised ms).

Lines 64-65: not only post-subduction geodynamic settings are characterised by an enormous flux of metals and volatiles, it also should be enriched in metals and volatiles in the ongoing-subduction arc.

We agree with the reviewer; however, the focus of this manuscript is on post-subduction processes.

Lines 73-78: This is not true, post-subduction magmatism is not generally comprised of <10% partial melting of subduction-metasomatised SCLM, on the contrast, widespread high Sr/Y and La/Yb ore-forming magmatism are derived from partial melting of lower crust.

Thanks for comment. As mentioned above, lower crust is not the sole source for Gangdese post-collisional magmatism. We have added a comment in this section that also acknowledges the crustal input (lines 269-302).

Line 114-118: should define the information about metasomatised mantle and some geochemical evidence of subduction-related metasomatism.

This is now addressed with the new samples; however, we now also cite evidence from references in lines 138-155.

Line 228-233: "We argue that low-degree partial melting of metasomatised SCLM during post-subduction magmatism is the trigger for transfer and concentration of metals and volatiles in

magmatic arcs..."this need more detailed evidence, Conversely, there more work showing that metals came from lower crust.

Again, we refer to Line 296-302 of revised ms.

Reviewers' comments:

Reviewer #2 (Remarks to the Author):

This manuscript has been significantly improved. However, there are still several significant problems the authors need to clear up.

1. The post-subduction magmatism is not always alkaline. Take the Gangdese belt (the best example of post-collisional settings) as the example, most post-subduction intrusions are high-K calc-alkaline. For this reason, I don't think this study can really build a global mode.
2. I am skeptical of the approach in this study. The rock samples selected from mantle, lower crust, mid-crust, and upper crust do not really constitute a continuum. The selected alkaline ultramafic pipes do not share a similar SCLM source with porphyries. For example, ultrapotassic volcanic rocks and high-Sr/Y granitoids in the Gangdese belt both developed in the Gangdese belt, and both concentrated in the mid-Miocene. However, they have huge differences in Sr-Nd-Hf-Pb-Os isotopes, suggesting they are coming from two different sources of SCLM and lower crust.
3. Although SCLM has been proposed for the source of post-subduction magmatism, it has never been a widely accepted model for the origin of high-Sr/Y granitic magmatism in the Gangdese belt. There are three reasons why we think SCLM is not the main source: 1) The ultrapotassic/potassic volcanic rocks derived from SCLM in the Gangdese belt have much lower ϵ_{Ndi} ratios (down to -20), suggesting the SCLM is quite isotopically evolved. However, all the high-Sr/Y granitoids have much higher ϵ_{Ndi} (up to +6) and ϵ_{Hf} ratios, which are similar to early subduction suites. This suggests those high-Sr/Y granitoids are derived from subduction-modified Tibetan lower crust, rather than SCLM. 2) If the granitic melts were mainly derived from SCLM, there should be mafic dykes around in the Gangdese belt. However, no mafic intrusions are discovered. 3) I would agree that hybridization of melts from SCLM and lower crust likely form the high-Sr/Y granitoids. However, the main source should still be the lower crust. Melts from SCLM just add a pinch of salt to the magmatic system.

Response to review:

NCOMMS-18-4584440A R2

“The metallogenic fingerprint of post-subduction magmatism”

By Holwell et al.

We appreciate the opportunity to revise and resubmit our manuscript to Nature Communications. We were pleased to hear that Reviewer 1 is happy with our revisions and recommends no further edits. Reviewer 2 states how the manuscript is significantly improved. However, he/she highlights three areas for further modification, which are all inter-related and pertain to broadly the same argument. We have taken on board these comments, and have addressed them; our responses are set out in bold in the revision notes below.

Reviewer #2 (Remarks to the Author):

This manuscript has been significantly improved. However, there are still several significant problems the authors need to clear up.

We are really pleased to hear that both reviewers have appreciated the revision efforts. We are grateful of the suggestions from the previous submission that have clearly helped to improve the manuscript.

1. The post-subduction magmatism is not always alkaline. Take the Gangdese belt (the best example of post-collisional settings) as the example, most post-subduction intrusions are high-K calc-alkaline. For this reason, I don't think this study can really build a global mode.

Many thanks for the comment. We totally agree that post-subduction magmatism goes from alkaline to calc-alkaline. We have reflected this range in compositions by referring to the magmatism as being 'alkali-enriched' following similar suggestions from both reviewers in the previous submission.

To appropriately represent the range of compositions we have revised our statement on this in the introductory sentences to help highlight this early on to be clear that the terminology in the remainder of the manuscript refers to 'alkali-enriched' and that this terms the broad range of compositions of magmas that are encountered in nature in these geodynamic settings. We now have the following statement in lines 49-51.

“This post-subduction process forms relatively small volume, hydrous magmas that range from high-K calc-alkaline, through silica-saturated to silica-undersaturated alkaline compositions¹, and are henceforth referred to as 'alkali-enriched'.”

We have gone through the manuscript and made four changes to ambiguous phrasing about the alkaline nature and now confirm that the use of 'alkali-enriched' is used throughout.

Furthermore, we do clearly state in lines 298-299 that the Gangdese rocks are generally more calc-alkaline, so we do not feel that we are misrepresenting the degree of alkalinity in that particular region. In fact, post-collisional magmatism in the Gangdese Belt ranges from alkaline including ultrapotassic volcanic rocks and lamprophyres (e.g. Gao et al., 2007, *Journal of Petrology*, v. 48, p. 729-752; Zhao et al., 2009, *Lithos*, v.113, p. 190-212; Sun et al., 2018, *Journal of Petrology*, v. 59, p. 341-386.), shoshonitic such as high-Mg diorite (Yang et al., 2015, *Journal of Petrology*, v. 56, p. 227-254.), medium-K calc-alkaline (e.g. mafic microgranular enclaves within ore-forming porphyries; Wu et al., 2014, *International Geology Review*, v. 56, p. 571-595; Lu et al., 2015, *Geology*, v. 43, p. 583-586.), to high-K calc-alkaline (e.g. high Sr/Y porphyries). The alkaline and shoshonitic mafic melts derived from metasomatized Tibetan mantle have been shown to be genetically related to the ore-forming high Sr/Y porphyries in the Gangdese belt (e.g. Yang et al., 2015, *Journal of Petrology*, v. 56, p. 227-254; Sun et al., 2018, *Journal of Petrology*, v. 59, p. 341-386.).

We have modified the text as follow (line 297 onward)

“Whereas the SCLM is the most likely source of metals in these systems, it can be argued that some may be sourced from the lower crust as well. For example, the generally more calc-alkaline post-subduction Gangdese porphyries have been suggested to be a product of melts from juvenile subduction-modified lower crust, based on their high Sr/Y and La/Yb ratios and unevolved isotopic compositions^{47,48}. However, several studies^{14,35,49} have clearly shown that lower crust melting alone cannot provide sufficient water to these systems (>10 wt%). Post subduction magmatism in the Gangdese belt ranges from ultrapotassic rocks¹⁰, through shoshonites³⁵ and medium K-calc alkaline¹⁴ to high K-calc-alkaline¹⁴ and the ore forming high Sr/Y porphyries have been shown to be genetically related to the alkaline and shoshonitic melts from metasomatized Tibetan mantle^{14, 35, 49}. Thus, it appears that although crustal melts may constitute the major source component in the Gangdese belt, the input of hydrous mantle melts from the metasomatised Tibetan lithospheric mantle is a small, but important requirement for the genesis of post-subduction porphyry mineralisation”

Therefore, we are convinced that this study can put forward a model that can be applied globally, identifying how these magmatic events and their related mineralisation may be part of a common post subduction, alkali-enriched thread.

2. I am skeptical of the approach in this study. The rock samples selected from mantle, lower crust, mid-crust, and upper crust do not really constitute a continuum. The selected alkaline ultramafic pipes do not share a similar SCLM source with porphyries. For example, ultrapotassic volcanic rocks and high-Sr/Y granitoids in the Gangdese belt both developed in the Gangdese belt, and both concentrated in the mid-Miocene. However, they have huge differences in Sr-Nd-Hf-Pb-Os isotopes, suggesting they are coming from two different sources of SCLM and lower crust.

We appreciate this comment and agree that we would have ideally preferred to examine mineralised crustal and metasomatised mantle rocks from the same locality. But given that natural laboratories satisfying this criterium are not available, we are convinced that the approach employed in this study is the best one can achieve. Furthermore, it is for the very reason that we use a dataset from a wide range of localities from different settings that we are able to demonstrate a common global thread.

As outlined in the previous answer, the revised manuscript recognises that the genesis of Miocene ore-forming porphyries in the Gangdese belt involves both SCLM and juvenile lower crust components with contrasting isotopic signatures (e.g. Yang et al., 2015, Journal of Petrology, v. 56, p. 227-254; Sun et al., 2018, Journal of Petrology, v. 59, p. 341-386.)

The rephrasing of the passage from line 297 as mentioned above in response to Point 1 covers this, specifically:

“Thus, it appears that although crustal melts may constitute the major source component in the Gangdese belt, the input of hydrous mantle melts from the metasomatised Tibetan lithospheric mantle is a small, but important requirement for the genesis of metal-rich post-subduction porphyry mineralisation.”

3. Although SCLM has been proposed for the source of post-subduction magmatism, it has never been a widely accepted model for the origin of high-Sr/Y granitic magmatism in the Gangdese belt. There are three reasons why we think SCLM is not the main source: 1) The ultrapotassic/potassic volcanic rocks derived from SCLM in the Gangdese belt have much lower ϵ_{Ndi} ratios (down to -20), suggesting the SCLM is quite isotopically evolved. However, all the high-Sr/Y granitoids have much higher ϵ_{Ndi} (up to +6) and ϵ_{Hf} ratios, which are similar to early subduction suites. This suggests those high-Sr/Y granitoids are derived from subduction-modified Tibetan lower crust, rather than SCLM. 2) If the granitic melts were mainly derived from SCLM, there should be mafic dykes around in the Gangdese belt. However, no mafic intrusions are discovered. 3) I would agree that hybridization of melts from SCLM and lower crust likely form the high-Sr/Y granitoids. However, the main source should still be the lower crust. Melts from SCLM just add a pinch of salt to the magmatic system.

We appreciate this comment. We agree that the main source of the high-Sr/Y granites in the Gangdese belt is juvenile subduction-modified Tibetan lower crust. The mineralized high-Sr/Y porphyries have additional input from hydrous mafic melts derived from SCLM such as those at Zhunuo and Qulong porphyry Cu deposit, the two most representative deposits from western and eastern Gangdese belt, respectively (Yang et al., 2015, Journal of Petrology, v. 56, p. 227-254; Sun et al., 2018, Journal of Petrology, v. 59, p. 341-386.). Given that pure lower crustal melts cannot provide enough water (>10 wt% H₂O) in the ore-forming Gangdese high Sr/Y porphyries, the input of SCLM melts are critical for adding water and possibly metals to the high Sr/Y melts to make the latter metallogenically fertile (Lu et al., 2015, Geology, v. 43, p. 583-586). This is the statement we make in the final line of the revised paragraph mentioned above in response to the other two comments (from line 297). We would argue that the ‘pinch of salt’ is actually the important part, and otherwise the melts would be rather insipid.

REVIEWERS' COMMENTS:

Dear Editor,

Thanks for sharing this new version to me. I can see the authors have made some changes, but I don't think they really address my comments.

I haven't see any solid evidence from their study and cited publications to prove that the mantle melts (ultrapotassic volcanic rocks) is an important requirement for the genesis of metal-rich post-subduction porphyry mineralization. I don't agree with their reply of "Given that pure lower crustal melts cannot provide enough water (>10 wt% H₂O) in the ore-forming Gangdese high Sr/Y porphyries, the input of SCLM melts are critical for adding water and possibly metals to the high Sr/Y melts to make the latter metallogenically fertile (Lu et al., 2015, *Geology*, v. 43, p. 583-586)."

1. Although high-Sr/Y granitoids in the Gangdese belt is likely water-rich (maybe >10 wt. % as Lu et al. 2015 proposed), there is no solid evidence that the SCLM melts provide water. This is just a hypothesis, no evidence. Those ultrapotassic volcanic rocks (UPVs) are thought to be derived from lithospheric mantle because of their evolved Sr-Nd-Hf isotopes. There is no evidence to prove that they are hydrous. Why are the melts from lithospheric mantle hydrous? Are there any magmatic water estimation from these rocks? How much water they can carry? In addition, except UPVs, subducted Indian continental plate can also provide water (Zheng et al. 2013)? Therefore, UPVs are not the necessary water source.

Zheng Y F, Zhao Z F, Chen Y X. Continental subduction channel processes: Plate interface interaction during continental collision. *Chin Sci Bull*, 2013, 58: 4371–4377, doi: 10.1007/s11434-013-6066-x

2. There is no evidence that the input of SCLM melts add metals to high Sr/Y melts to make the system fertile. Most importantly, Cline and Bodnar (1991) showed that large porphyry Cu deposits can readily be formed from normal arc magmas containing only 50 ppm Cu.

Cline, J.S., and Bodnar, R.J., 1991, Can economic porphyry copper mineralization be generated by a typical calc-alkaline melt? *Journal of Geophysical Research*, v. 96, p. 8113–8126.

3. The genetic model should be addressed. Partial melting of lower crust should be highlighted in your model.

Response to review:

NCOMMS-18-4584440A R3

“The metallogenic fingerprint of post-subduction magmatism”

By Holwell et al.

Following further revision, Reviewer 2 still had repeated comments with regards to the interpretation of the Gangdese belt post subduction magmatism.

In this case, and following the offer from the editor of the opportunity to publish what we have, and invite a Comment piece from Reviewer 2, we have decided to make no further changes with regards to how we have described the Gangdese belt case study area. We feel that we have acknowledged in the manuscript that there is controversy over the origin of these magmas, but that to go further into addressing the reviewer’s comments we would imbalance the manuscript and also advocate a preferred interpretation that we do not necessarily share with the reviewer.

As such, we respectfully note the comments below but have not revised our manuscript further.

Reviewer #2 COMMENTS:

Dear Editor,

Thanks for sharing this new version to me. I can see the authors have made some changes, but I don’t think they really address my comments.

I haven’ see any solid evidence from their study and cited publications to prove that the mantle melts (ultrapotassic volcanic rocks) is an important requirement for the genesis of metal-rich post-subduction porphyry mineralization. I don’t agree with their reply of “Given that pure lower crustal melts cannot provide enough water (>10 wt% H₂O) in the ore-forming Gangdese high Sr/Y porphyries, the input of SCLM melts are critical for adding water and possibly metals to the high Sr/Y melts to make the latter metallogenically fertile (Lu et al., 2015, *Geology*, v. 43, p. 583-586).”

1. Although high-Sr/Y granitoids in the Gangdese belt is likely water-rich (maybe >10 wt. % as Lu et al. 2015 proposed), there is no solid evidence that the SCLM melts provide water. This is just a hypothesis, no evidence. Those ultrapotassic volcanic rocks (UPVs) are thought to be derived from lithospheric mantle because of their evolved Sr-Nd-Hf isotopes. There is no evidence to prove that they are hydrous. Why are the melts from lithospheric mantle hydrous? Are there any magmatic water estimation from these rocks? How much water they can carry? In addition, except UPVs, subducted Indian continental plate can also provide water (Zheng et al. 2013)? Therefore, UPVs are not the necessary water source.

Zheng Y F, Zhao Z F, Chen Y X. Continental subduction channel processes: Plate interface interaction during continental collision. *Chin Sci Bull*, 2013, 58: 4371–4377, doi: 10.1007/s11434-013-6066-x

2. There is no evidence that the input of SCLM melts add metals to high Sr/Y melts to make the system fertile. Most importantly, Cline and Bodnar (1991) showed that large porphyry Cu deposits can readily be formed from normal arc magmas containing only 50 ppm Cu.

Cline, J.S., and Bodnar, R.J., 1991, Can economic porphyry copper mineralization be generated by a typical calc-alkaline melt? *Journal of Geophysical Research*, v. 96, p. 8113–8126.

3. The genetic model should be addressed. Partial melting of lower crust should be highlighted in your model.